# Obesity Prevention within the Early Childhood Education and Care Setting: A Systematic Review of Dietary Behavior and Physical Activity Policies and Guidelines in High Income Countries

**DOI:** 10.3390/ijerph18020838

**Published:** 2021-01-19

**Authors:** Jacklyn Kay Jackson, Jannah Jones, Hanh Nguyen, Isabella Davies, Melanie Lum, Alice Grady, Sze Lin Yoong

**Affiliations:** 1Faculty of Health and Medicine, School of Medicine and Public Health, University of Newcastle, Callaghan, NSW 2308, Australia; Jacklyn.Jackson@health.nsw.gov.au (J.K.J.); Jannah.Jones@health.nsw.gov.au (J.J.); Melanie.Lum@health.nsw.gov.au (M.L.); Alice.Grady@health.nsw.gov.au (A.G.); 2Hunter Medical Research Institute (HMRI), New Lambton, NSW 2305, Australia; 3Priority Research Centre for Health Behavior, School of Medicine and Public Health, University of Newcastle, Callaghan, NSW 2308, Australia; 4National Centre of Implementation Science (NCOIS), School of Medicine and Public Health, University of Newcastle, Callaghan, NSW 2308, Australia; 5Hunter New England Population Health, Hunter New England Local Health District, Wallsend, NSW 2305, Australia; 6School of Health Sciences, Swinburne University of Technology, Hawthorn, VIC 3122, Australia; 102486563@student.swin.edu.au (H.N.); 102463207@student.swin.edu.au (I.D.)

**Keywords:** early childhood education and care, obesity prevention, policies and practices, dietary behavior, physical activity, systematic review, practice guidelines

## Abstract

As a strategy for early childhood obesity prevention, a variety of dietary behavior and physical activity policies and guidelines published by leading health agencies and early childhood education and care (ECEC) licensing and accreditation bodies exist. Given the potential diversity in recommendations from these policies, this narrative review sought to synthesize, appraise and describe the various policies and guidelines made by organizational and professional bodies to highlight consistent recommendations and identify opportunities to strengthen such policies. An electronic bibliographic search of seven online databases and grey literature sources was undertaken. Records were included if they were policies or guidelines with specific recommendations addressing dietary behavior and/or physical activity practice implementation within the ECEC setting; included children aged >12 months and <6 years and were developed for high income countries. Recommended dietary behavior and physical activity policies and practices were synthesized into broad themes using the Analysis Grid for Environments Linked to Obesity framework, and the quality of included guidelines appraised. Our search identified 38 eligible publications mostly from the US and Australia. Identified guidelines were largely consistent in their recommendation and frequently addressed the physical and sociocultural environment and were well-aligned with research evidence. Broader consideration of policy and economic environments may be needed to increase the impact of such policies and guidelines within the ECEC setting.

## 1. Introduction

Childhood overweight and obesity is increasingly prevalent, and if global trends continue, will affect up to 70 million infants and young children by the year 2025 [1]. Childhood obesity is defined by an excess of body fatness that is widely categorized according to body mass index scores adjusted for child sex and age [2]. Given that childhood obesity can track throughout the lifespan and influence lifelong health trajectories [3], it has been identified as one of the most serious public health challenges of the 21st century [1]. As such, population-based obesity prevention strategies including policy, practice and environmental strategies to support dietary behaviors and physical activity in infants and young children are considered a key public health approach for addressing the global childhood obesity epidemic [4].

Early childhood education and care (ECEC) is increasingly acknowledged as a setting that can offer the foundations for lifelong child learning and development [5]. ECEC services are inclusive of regulated care services such as long day care, preschools, nurseries, kindergartens, and occasional care services that cater for children, prior to attending compulsory schooling [6]. Given that ECEC utilization tends to be higher within communities of greater socioeconomic advantage and with a higher degree of tertiary educated mothers, across high income Organization for Economic Co-operation and Development (OECD) member countries, approximately 50% to 90% of children aged 0–2 years and 3–5 years, respectively, attend some form of ECEC for on average 25–35 h a week [7]. Thus, given that the ECEC setting provides access to a large number of young children for prolonged and regular periods of time at a foundational period in life, it represents a key setting for the delivery of public health obesity prevention strategies targeting dietary behaviors and physical activity [8].

The importance of creating supportive environments to improve dietary behaviors and physical activity has long been formally recognized [9], and is a notion that is reinforced by a variety of socio-ecological models highlighting the interdependence between individual behavior, their health and their environment [10]. One such model is the Analysis Grid for Environments Linked to Obesity (ANGELO) framework [11], which suggests that within micro-environments such as the ECEC setting, dietary behaviors and physical activity are influenced by the physical (i.e., what is available), economic (i.e., the cost of the behavior), political (i.e., the rules) and sociocultural (i.e., attitudes, beliefs, and values) environments of the setting [11]. As such, comprehensive community-based obesity prevention strategies within the ECEC sectors should seek to target a combination of these factors [10]. An example of an ECEC strategy addressing the physical environment may relate to ensuring only nutritious foods and beverages are offered to children whilst in care. An example of how a strategy could address the economic environment of the ECEC may relate to offering subsided healthy meals to children in the setting. A strategy directly aligned with the policy environment would relate to ECEC services implementing a policy outlining explicit details around how the service will commit to improving child dietary behaviors and physical activity. Strategies addressing the sociocultural environment may relate to educators’ role modeling healthy dietary behaviors and physical activity.

Practice guidelines are a useful tool for assisting ECEC services to identify and implement appropriate practices to support healthy lifestyle behaviors in the setting. Such guidelines can offer an overview of current best practice evidence and recommendations, developed in consultation with end-users to contextualize empirical evidence for implementation [12]. As such, there are a variety of best practice guidelines as well as standards for licensing and accreditation that recommend ECEC services implement policies and practices to improve the quality of child diet and increase the time children spend being physically active in care, while also reducing sedentary behavior [13,14,15]. These sector specific guidelines are frequently produced by different organizations and professional bodies, and although likely have substantial overlap in scope, may produce varying recommendations which could result in confusion for the sector, as well as practitioners tasked with supporting policy implementation [16,17,18,19]. Synthesis of the existing policies and guidelines identifying the potential commonalities and differences in recommendations, together with an assessment of methodological quality, can provide useful insights in regards to areas of broad coverage, as well as highlight opportunities to strengthen current guidelines. However, the structural, organizational and funding models applied to the ECEC setting can vary substantially between high and low income countries, which is likely to influence the available infrastructure and political support for obesity prevention initiatives in this setting [20]. For this reason, this review sought to examine only the policies and guidelines in high income countries to ensure context consistency, comparability and relevance. Therefore, the aim of this systematic review is to: (1) identify ECEC based obesity prevention policies and guidelines in high income OECD countries; (2) assess the methodological quality of these policies and guidelines; and (3) synthesize the recommended obesity prevention policies and practices for implementation in the ECEC setting.

## 2. Materials and Methods

This review has been conducted and reported in accordance to the PRISMA guidelines [21]. A protocol or registration of this review has not been previously published.

### 2.1. Inclusion and Exclusion Criteria

This review sought to include ECEC focused recommendations and guidelines that aimed to prevent obesity in preschool aged children and targeted specific obesity prevention behaviors including dietary behaviors (inclusive of guidelines influencing food choices (i.e., preferences and intentions), eating behaviors (i.e., eating habits, eating occasions and portion sizes), and dietary intake/nutrition (i.e., food and nutritional make-up on intakes and diets) [22]), physical activity, sedentary behavior/screen time and/or sleep.

For the purposes of this review, policies and guidelines were defined as a written document containing practice and policy focused recommendations, and were developed and endorsed by an authoritative, professional or expert body/organization. Given the wide variation in cultural, social, ecological and political contexts internationally (which can influence how communities and governments prioritize early childhood obesity prevention) [20,23], this review focused on both regional and national guidelines from countries ranked within the OECD top 20 countries for highest average annual incomes for 2019 [24]. These countries included Iceland, Luxembourg, Switzerland, United States, Denmark, Netherlands, Belgium, Australia, Norway, Austria, Germany, Canada, Ireland, United Kingdom, Sweden, France, Finland, New Zealand, South Korea and Slovenia. Guidelines were eligible for inclusion in the review if they: (1) Promoted obesity prevention policies or practice implementation within the ECEC setting (this included regulated care services such as long day care, preschools, nurseries, kindergartens, occasional care services that cater for children prior to attending compulsory schooling [6]). Included guidelines could promote obesity prevention policies or practices broadly, or focus on specific obesity prevention behaviors including dietary behavior, physical activity, sedentary behavior, screen-time or sleep; (2) were aimed at improving obesity prevention behaviors of preschool children aged greater than 12 months and under 6 years old [7]; (3) were developed for contexts relevant to high income OECD countries; and (4) were published in English language.

ECEC based practices developed as part of obesity prevention interventions/programs and evaluated in conventional trials (e.g., Nutrition and Physical Activity Self-Assessment for Child Care (NAPSACC)) [25], were not included in this definition. We also excluded any setting-based licensing and accreditation criteria documents, which provide only broad overviews for the minimum standards of practice. Additionally, we excluded documents focused on dietary behavior and physical activity in young children more broadly, without describing ECEC specific recommendations, and did not focus on a particular jurisdiction (e.g., World Health Organization: Guidelines on physical activity, sedentary behavior and sleep for children under 5 years of age [26]).

If one advisory body or organization had published more than one guideline on the same subject, then the most recent update was selected for inclusion. Thus, only findings from the latest version of published guidelines were included for synthesis.

### 2.2. Search Strategy

To identify ECEC based obesity prevention policies and guidelines, an online search of electronic bibliographic databases including Medline, CINAHL complete and ClinicalKey was conducted, from inception until September 2020. A search of online guideline databases including the National Institutes of Health (https://www.nhi.gov/health-information); World Health Organization (https://www.who.int); National Heart, Lung, Blood Institute (NHLBI) Publication and Resources (https://www.nhlbi.nih.gov/health-topics/all-publications-and-resources); and Guideline International Network (https://g-i-n.net) was also conducted. The search combined the following search terms: (1) ECEC and, (2) Diet or physical activity or screen time or sedentary behavior or sleep or obesity, and (3) Policy/guidelines (the Medline Search Strategy is available in Appendix A).

In addition, relevant grey literature sources including Google and Google Scholar were searched (first 50 pages of results), as well as various websites of National ECEC quality bodies including: Australian Children’s Education & Care Quality Authority (ACECQA); Childcare resource and research unit (Toronto, ON, Canada); Office of Child Care (Washington, DC, USA); CARE—European Early Childhood Education and Care; Ministry of Education (Wellington, New Zealand); Department of Health, Social Services and Public Safety (Belfast, UK); and Korea Child Care Promotion Institute (Grey Literature Search Strategy available in Appendix A). Any relevant articles or reports identified through these searches underwent hand reference list checks by an independent reviewer (J.K.J) to identify any additional guidelines that may have been missed.

### 2.3. Selection of Guidelines

Once duplicates were removed, all identified titles and abstracts were screened by a single reviewer against the inclusion criteria in Endnote (J.K.J., H.N. and I.D). Titles and abstracts that were not deemed relevant were excluded from further review. Records selected for full-text screening were assessed against the inclusion and exclusion criteria independently by H.N. and I.D., with any uncertainties resolved by consensus or by a third reviewer (J.K.J). Reasons for excluding full-text records were documented in a Microsoft Excel spread sheet and reported in a PRISMA diagram [21] (See Figure 1).

### 2.4. Data Extraction

Three reviewers (J.K.J., H.N. and I.D.) extracted data from included policies and guidelines to a Microsoft Excel spreadsheet. Data related to the name of the guideline, the guideline jurisdiction, publishing body, publication year, and target group were extracted. The specific details of the recommended obesity prevention policies and practices were also extracted and grouped according to the targeted obesity prevention behavioral risk factors (i.e., dietary behavior; physical activity, sedentary behavior, screen time or sleep).

### 2.5. Assessment of Methodological Quality

To assess the methodological quality of the included guidelines, the Appraisal of Guidelines for Research & Evaluation II (AGREE II) tool was used [27]. The AGREE II tool was developed to assess the quality of healthcare guidelines developed by local, national or international groups and affiliated governmental organizations [27], and was used to assess the included guidelines across the following six domains:(1)Scope and Purpose: this relates to the overall aim of the guideline, and the target population (2 items).(2)Stakeholder Involvement: focuses on the extent and reporting to which appropriate stakeholders and views of its intended users were considered during the development of guidelines (3 items).(3)Rigor of Development: relates to the processes used to gather and synthesize the evidenced used to formulate the recommendations (8 items).(4)Clarity of Presentation: deals with the language structure, and format of the guideline (2 items).(5)Applicability: relates to consideration of barriers and facilitators to implementation, strategies to improve uptake, and availability of resources to apply the guidelines (4 items).(6)Editorial Independence: this considers if the formulation of the guidelines are not being unduly biased by funding agencies, and reporting of competing interests (2 items).

Each item within the tool was rated on a 7-point scale (1 = strongly disagree; 7 = strongly agree). Methodological assessment of included guidelines were conducted by review authors (H.N. and I.D.). A sub-sample (50%) were checked by a second review author (M.L. or A.G.). A quality score for each of the six domains was calculated by summing the score awarded across all domain items, then scaling the total score as a percentage (using the maximum possible score for that domain as the denominator). As specified in the AGREE II user-manual, a final judgment score was also awarded to the overall quality of the guideline, taking into account all criteria assessed [27]. While AGREE II does not specify a specific approach for reaching an overall quality score, reviews of the AGREE II assessments have indicated that domain 3 (rigor of development) is the strongest predictor of guideline quality [28]. Thus using methods aligned with previous reviews of guidelines [29], we considered a guideline high in quality (and recommended without modification) if it achieved a score of >60% for rigor of development (domain 3), as well as >60% in any two other domains. If guidelines scored <60% across all domains, this guideline was considered low in quality and not recommended.

### 2.6. Data Synthesis and Analysis

Recommended healthy eating and physical activity policies and practices within guidelines were qualitatively synthesized into broad policy/practice themes and sub-themes. To ascertain the consistency and frequency in which certain policy/practice items were recommended, we conducted a qualitative synthesis of the extracted items to identify overarching themes and subthemes using a deductive approach similar to other reviews of effective interventions in the ECEC setting [30,31,32]. The frequency in which identified overarching themes and sub-themes were recommended across the included guidelines was coded by an independent reviewer (J.K.J), and confirmed with a second reviewer (S.L.Y) with extensive experience in the setting to reach consensus on the final list of policy and practice themes. The identified themes were then mapped to the ANGELO framework category definitions to provide an overview of where the primary recommendations focus and to guide our narrative summary of study results [11]. The ANGELO framework is appropriate to apply to microenvironments such as the ECEC setting [11], and groups environmental strategies into four categories: (1) Physical (i.e., addresses what is available); (2) Economic (i.e., addresses the costs and economic influences); (3) Political/Policy (i.e., refers to the rules, laws, regulations and policies); and 4) Sociocultural (i.e., refers to attitudes, beliefs, values and social and cultural norms) [11].

## 3. Results

### 3.1. Guideline Selection

The PRISMA flow diagram of guideline selection is shown in Figure 1. The electronic database, online guideline database and grey literature searches resulted in a total of 7777 records. Once duplicates were removed, 5710 titles and abstracts were screened. A total of 108 full-text articles were screened against review eligibility criteria, of which 38 records were identifed as eligible for final inclusion in the review.

### 3.2. Guideline Characteristics

The characteristics of all 38 included guidelines can be found in Table 1. Twenty included guidelines were specifically developed for the US context, nine for Australia, five for the UK, two for New Zealand, and one each for Canada and Ireland. A total of 14 guidelines were developed at a National level, and 24 were developed specifically at a regional/state level.

The majority of guidelines included ECEC policy and practice recommendations targeting both dietary and physical activity risk behaviors (*n* = 21), 10 focused solely on dietary behavior policies and practices [33,34,35,36,37,38,39,40,41], and seven focused on physical activity policies and practices (including sedentary behavior, screen time and sleep recommendations) [17,42,43,44,45,46,47]. The publication year of included guidelines ranged from 1999–2020, and 19 out of 38 included guidelines were published within the past 5 years. 

### 3.3. Quality of Guidelines

The domain standardized AGREE II scores for each included guideline and their corresponding overall quality recommendation is presented in Appendix A.

Overall, guidelines scored highest (average score 93%) in domain 1 (Scope and purpose), with 21 out of 38 guidelines scoring 100%, for clearly specifying the objective and target population of the guideline.

Included guidelines generally scored moderately (average score 72%) for domain 2 (Stakeholder involvement). The highest score for this domain was 100% (*n* = 2) [45,65], however 11 guidelines scored ≤60% for this domain, with explicit details on the group that contributed to the developed the guidelines and/or the consideration of the target group preferences often missing.

Included guidelines scored poorly (average score 47%) for domain 3 (Rigor of development), with only three [45,51,53] guidelines reporting adequate detail to score >60% across this domain.

For domain 4 (Clarity of presentation), included guidelines scored well (average score 85%), with many guidelines providing easy to identify, and specific recommendations.

On average included guidelines scored moderately (average score 64%) for domain 5 (Applicability), with many guidelines providing tools and resources to support guideline implementation, however consideration of the barriers, facilitators and resource implications to guideline implementation were less clearly addressed overall.

For domain 6 (Editorial independence) guidelines scored only moderately (average score 57%), with explicit details related to guideline development funding sources or conflict of interests rarely explicitly addressed.

Finally, based on the overall quality assessment scores, it was determined that three of the 38 included guidelines were recommended without modification [45,51,53], 34 were recommended with modification, and one guideline [49] was not recommended (Table 1).

### 3.4. Dietary Behavior Policy and Practice Recommendations Grouped according to the ANGELO Framework

Recommended dietary behavior policies and practices are summarized in Table 2. While recommended dietary behavior practices frequently addressed the physical and sociocultural environment of the ECEC setting, very few addressed the policy environment, and none addressed the economic environment.

#### 3.4.1. Dietary Behavior Recommendations addressing the Physical Environment

The most common dietary behavior recommendation was directed at modifying the physical environment of the ECEC setting, with 30 dietary behavior guidelines recommending nutrition standards for the food and beverages offered. The most common recommendation was to ensure foods and beverages served within the ECEC setting align with National nutrition guidelines for the jurisdiction, and providing a variety of healthy foods from the main food groups in age appropriate portion sizes (*n* = 28), followed by recommendations to limit the serves and types of sugar sweetened beverages (including fruit juice) offered (*n* = 25). Ensuring that water is widely available to children at all times (*n* = 23) and offering age-appropriate milks and beverages (*n* = 22) were also frequently recommended. It was also a common (*n* = 10) recommendation to keep high energy, low nutrient foods out of the ECEC setting.

Twenty five guidelines made policy/practice recommendations related to creating an environment that encourages and promotes healthy dietary behaviors. More specifically, guidelines frequently recommended that ECEC services provide healthy options at meal- times in age appropriate serve sizes and allow children to self-serve (i.e., choose which foods and how much) (*n* = 14). Creating a relaxed, enjoyable and social meal-time environment, that encourages child-to-child and child-to-educator interactions was also frequently recommended (*n* = 12). Additionally, recommendations to ensure meal and snack times are scheduled in a consistent and predicable manner (*n* = 10), and allowing children adequate time to eat (*n* = 9) were also common. The explicit recommendation to ensure fruit and vegetable snacks are widely available and easily accessible within the ECEC setting was only recommended within one guideline (*n* = 1).

#### 3.4.2. Dietary Behavior Recommendations Addressing the Sociocultural Environment

Many guidelines (*n* = 23) targeted the sociocultural environment to influence dietary behaviors in the ECEC setting. This included recommendations related to ECEC educator feeding practices to encourage healthy dietary behaviors (*n* = 16). Specifically, seven guidelines recommended that educators should encourage children to taste different fruits and vegetables each day (and praise them when they do), and five recommended that educators sit with the children at meal-times and role model healthy eating behaviors (i.e., eat and drink healthy foods and beverages in front of children). The recommendation that food should not be used as a reward or punishment was referenced in nine guidelines. However, only three guidelines explicitly recommended that children should not be forced to eat or bribed with food. Further, the recommendation to avoid celebrating special occasions with food or using food as a reward was only explicitly recommended in two guidelines. Other less common recommendations related to ensuring educators are trained in delivering nutrition curriculum and practices (*n* = 4).

Of the guidelines that made recommendations related to the ECEC providing nutrition education (*n* = 11), all specified recommendations focused on ensuring the ECEC offer children the opportunity to engage in a variety of food awareness and education activities (to allow children to experiment with different foods, shapes, colors and textures, as well as discuss food preferences and food traditions).

Of the nine guidelines that recommended some form of parent engagement strategy, six of the guidelines recommended that parents are engaged to ensure foods packed from home are healthy and meet nutrition standards. Six guidelines also recommended that parents are engaged to make suggestions to the development of a healthy ECEC menu. Other parent engagement strategies such as providing parents with a copy of the ECEC written nutrition guidelines, providing a copy of the ECEC menu, and offering parent nutrition education were less frequently recommended.

#### 3.4.3. Dietary Behavior Recommendations Addressing the Policy Environment

The explicit recommendation that ECEC services should develop and implement a nutrition or healthy eating policy was rarely addressed within included guidelines (*n* = 3). However, of the three that did, it was recommended that nutrition policies are reviewed annually by a nutrition expert, are developed in consultation with parents and staff, and widely disseminated to relevant stakeholders (i.e., parents and staff).

### 3.5. Physical Activity Policy and Practice Recommendations Grouped According to the ANGELO Framework

Recommended physical activity, sedentary behavior, screen time and sleep policies/practices are presented in Table 3. Recommendations for physical activity largely addressed the physical and sociocultural environment. The policy environment was less frequently addressed, and no guidelines addressed the economic environment.

#### 3.5.1. Physical Activity Recommendations Addressing the Physical Environment

Out of the 28 included guidelines that provided recommendations for physical activity, all included a broad recommendation for ECEC services to provide opportunities for children to be physically active, indicating that more physical activity is better. Of these, 16 recommended that at least 180 min of physical activity of any intensity should be offered throughout the day, and 12 recommended that children aged 3–4 years should participate in 60 min of moderate-to-vigorous physical activity during the day. Eleven guidelines recommended that ECEC services should provide daily opportunities for child physical activity by providing supervised outdoor playtime. Ten guidelines recommended that ECEC services provide structured, adult-led physical activity opportunities, and nine recommended services offer unstructured, free-play physical activity opportunities. Only five guidelines made explicit recommendations for ECEC services to ensure physical activity is incorporated into the formal curriculum and daily routine.

The recommendation to create a physical environment that promotes physical activity was included in six guidelines. Specific recommendations for supporting this included providing play equipment that encourages physical activity (*n* = 4), providing simple portable play equipment that encourages creative play indoors and outdoors (*n* = 4), ensuring the ECEC service provides adequate space to be physically active (*n* = 4), and ensuring the outdoor area offers a variety of secure equipment under shade, as well as varied play surfaces and open grass areas (*n* = 2).

Twenty three included guidelines recommended that ECEC services limit the use of screen time. Eighteen of these explicitly recommended that children aged 2 or above should engage in no more than 1 h of screen time a week, and 16 recommended no screen time for children aged under 2 years. Five guidelines recommended that screen use is limited for educational or active movement activities/programs, and four recommended that when offered, screen/digital media should be free from advertising, violence or sounds/colors that will tempt children to overuse. Only two guidelines explicitly recommended that screen time should be supervised by an adult, and only one guidelines stated that parent permission is requested for children to participate in any screen based activity. The recommendation to limit the time children spend siting was made in 14 guidelines, and 12 of these recommended that children should not be sitting or restrained for more than one hour at a time.

Healthy sleeping habits and practices were recommended in nine of the included guidelines. Six guidelines recommended that a nap be embedded within the daily routine (with regular sleep and wake-up times). Sleeping recommendations also related to providing a calm nap time routine and providing an environment that is quiet and screen free.

#### 3.5.2. Physical Activity Recommendations Addressing the Sociocultural Environment

Eight included guidelines recommended that educators promote the benefits of physical activity with the children. More specifically, guidelines recommend educators role model physical activity by participating in activities (*n* = 5), engage children in fun physical activity including games and sports they will enjoy (*n* = 5), avoid withholding physical activity as a punishment (*n* = 4), engage children in expressive play (e.g., music, dancing and make believe) (*n* = 3), and educators embed physical activity into educational activities (*n* = 2). Four included guidelines recommended that when children are sedentary, they should be engaged in educational, creative or social pursuits. One guideline explicitly recommended that educators should avoid punishing children for being physically active.

The least common recommendation across included guidelines related to ECEC services offering educator training to provide safe and developmentally appropriate physical activity (*n* = 2), which made specific recommendations around ensuring staff and parents were adequately educated and trained to provide appropriate quantities and types of physical activity, sedentary behavior and sleep.

#### 3.5.3. Physical Activity Recommendations Addressing the Policy Environment

Three guidelines recommended that ECEC services should adopt standards for physical activity and physical activity education programs. Only one guideline included specific recommendations for services to engage staff and parents to support ECEC physical activity standards. Recommendations to seek consultation from experts annually on physical activity programs delivered in ECEC; to provide parent education; and develop a written policy promoting physical activity and removing barriers to physical activity participation were recommendations specified in one guideline each.

## 4. Discussion

To our knowledge, this is the first attempt to systematically synthesize findings from guidelines and policies for obesity prevention in ECEC settings from high income countries. This review provides an overview of where the most overlap in recommendations are and found large consistencies in recommendations that targeted the physical environment of the ANGELO framework. The recommendations related to sociocultural and policy environments were less consistent and few policies addressed these domains. As the guideline development process typically relies on both the existence of relevant empirical evidence as well as expert opinion to contextualize findings, minor variations in recommended policies and practices were expected, given guidelines are likely to be tailored to a particular jurisdiction. We found 38 guidelines overall, with the majority published to provide guidance for various regions of the US (*n* = 18), Australia (*n* = 7) and UK (*n* = 5). Such findings are consistent with where the majority of primary intervention research targeting childhood obesity prevention in the ECEC setting has been undertaken [70,71].

In order for ECEC guidelines and policies to have an impact on child health outcomes, they need to be supported by empirical evidence. Our review identified that the most consistent recommendations were strongly supported by empirical evidence. For dietary behavior, the most prevalent recommendation centered on food availability within the setting (*n* = 30). This included recommendations that food and beverages served within the ECEC setting align with National dietary guidelines, recommendations on limiting sugar sweetened beverages and providing healthy options at meal times in age-appropriate serve sizes. Such findings are consistent with randomized controlled trial evidence suggesting that increasing the provision of foods consistent with nutrition guidelines [72,73,74], and restricting availability of sugar sweetened beverages in ECEC settings are effective in improving child diet [75].

A smaller number of dietary behavior guidelines included recommendations around creating relaxed, enjoyable and social meal-time environments and those targeting educator-child interactions. Although the evidence for this practice within ECEC specifically is limited [76], there is strong evidence that such practices in the home and other environments may be useful in increasing the selection and consumption of fruit and vegetables among children aged <5 years [77]. Surprisingly, less than 1/3rd of dietary behavior guidelines recommended some form of parent engagement strategy. Despite mixed evidence surrounding the impact of parent involvement on child diet [78,79], there is some suggestion that involvement of parents particularly around the packing of childcare lunchboxes can improve vegetable intake [80].

For physical activity, all guidelines recommended providing opportunities to be physically active, with over half specifying at least 180 min which is consistent with that outlined in international guidelines by WHO [26]. Additionally, 23 guidelines included recommendations around the use of screen time, also consistent with the duration recommended by WHO [26]. Less than half of included guidelines provided recommendations related to the offering of structured/supervised play, as well as free play and outdoor free play opportunities. Only six guidelines provided recommendations on creating a physical environment that promotes physical activity despite consistent empirical evidence indicating the benefits of such interventions on child physical activity [32,81], suggesting there may be opportunity to strengthen the impact of such guidelines by the inclusion of such recommendations.

Guided by the ANGELO framework, we identified few guidelines that addressed the policy environment and none addressed the economic environment, findings similar to that documented in a meta-review of evidence assessing the impact of fruit and vegetable interventions [82]. While service-level policies as a standalone strategy may not be effective in improving child diet or activity [30], nutrition and physical activity policies are important tools to communicate center priorities, help guide the practice of ECEC settings and can be useful to provide clear and consistent information to both staff and parents. This points to an opportunity for future guidelines to provide clear guidance around the development and implementation of policies and to consider how economic environments may influence the ability of an ECEC service to influence child dietary behavior, physical activity and sedentary behavior. Additionally, there were few recommendations around tailoring dietary behavior and physical activity opportunities to be more culturally appropriate. Given the broad reach and diversity of children attending such settings, future guidelines should consider the inclusion of culturally competent guidance, as cultural and religious preferences are increasingly acknowledged as an important mediator for child access, acceptability and preferences for certain health behaviors known to influence child health outcomes [83].

It was reassuring to find that ECEC based obesity prevention policies and guidelines aligned with current research evidence but highlights the need for ongoing investment to support policy implementation. Even with the introduction and existence of such policies in a number of countries in the last decade, alarmingly the rates of childhood obesity have continued to increase [84,85]. While many studies have shown that despite the availability of such guidance, few ECEC services implement these evidenced-based recommendations [72,86]. Thus, while there remain opportunities to strengthen current guidelines, the impact of such policies cannot be achieved without the systematic development of dissemination and implementation strategies to facilitate their successful update [87,88]. Additionally, the introduction of macro-level incentives and strategies such as their inclusion in regulatory standards, may be useful to facilitate the wide-spread adoption of such recommendations within the setting.

Surprisingly, we only identified guidelines from six of the 20 countries included in this review, highlighting a possible need for specific guidance for the ECEC sector in these countries to support obesity prevention efforts. Additionally, for many jurisdictions, the guidelines were published over 10 years ago, which is likely to warrant an update to capture progression of research evidence in the field. Our assessment of guideline quality using the AGREE-2-tool identified three guidelines that scored highly on the tool and were recommended without modification. Thus, given that very few included guidelines scored high quality ratings, our synthesis has included findings from all identified policies and guidelines regardless of quality, as exclusion of guidelines due to low quality ratings would have precluded the inclusion of recommendations from a number of countries. However, it is possible that the guidelines identified as high quality could be adopted or be used as a reference document for jurisdictions that do not have existing ECEC guidelines, provided they are contextually relevant and appropriate. Additionally, findings from this review which provide an overview of the commonalities of guidelines, may be useful in future efforts to develop unified, overarching guidelines providing consistent advice for countries where there are similarities in political and governance structures for ECEC. Further, although the existence of these recommendations are useful to guide practice, there is a need for investment in implementation support to ensure such policies are translated into routine ECEC-practice in order to increase child health.

### Limitations

This study needs to be interpreted in light of several study limitations. Although we undertook a systematic search of electronic databases, online guidelines and grey literature searches, identifying guideline documents is a challenging task, with limited guidance available for conducting such a search. We also excluded non-English language publications in this review, which may have resulted in fewer countries included in this assessment, as such, it is possible that we may have missed some relevant guidelines. Additionally, we only included documents that had a clear focus on recommendation within the ECEC. Given this, we may have missed recommendations from policies that focused on child health more broadly but also included ECEC recommendations within the text. The AGREE-II assessment was undertaken using only information that was freely available online, as such guidelines were scored poorly if the information was unavailable. We undertook a narrative synthesis to provide a high-level summary of frequency in recommendations as this was the primary purpose of the review. Despite such limitations, to our knowledge this represents the first systematic process to summarize the recommendations from obesity prevention guidelines in the ECEC setting in high-income countries. While the inclusion of guidelines in this review may not be exhaustive, we applied systematic methodology to identify a representative sample of setting guidelines. As such, this review provides an overview of the types of overarching practices that may warrant implementation, as well as highlights opportunities to strengthen current guidelines.

## 5. Conclusions

This review identified recommendations that ECEC services can implement to support the development of dietary and physical activity behaviors in children attending such settings. The guidelines summarized here were mainly from the US and Australia. The majority of recommendations focus on the physical and sociocultural environment, highlighting opportunities to better address the policy and economic environment. This review provides an overview of the recommendations and identifies high quality ECEC based policies and guidelines could form as the basis for developing future childhood obesity prevention guidance or unified recommendations for the sector.

## Figures and Tables

**Figure 1 ijerph-18-00838-f001:**
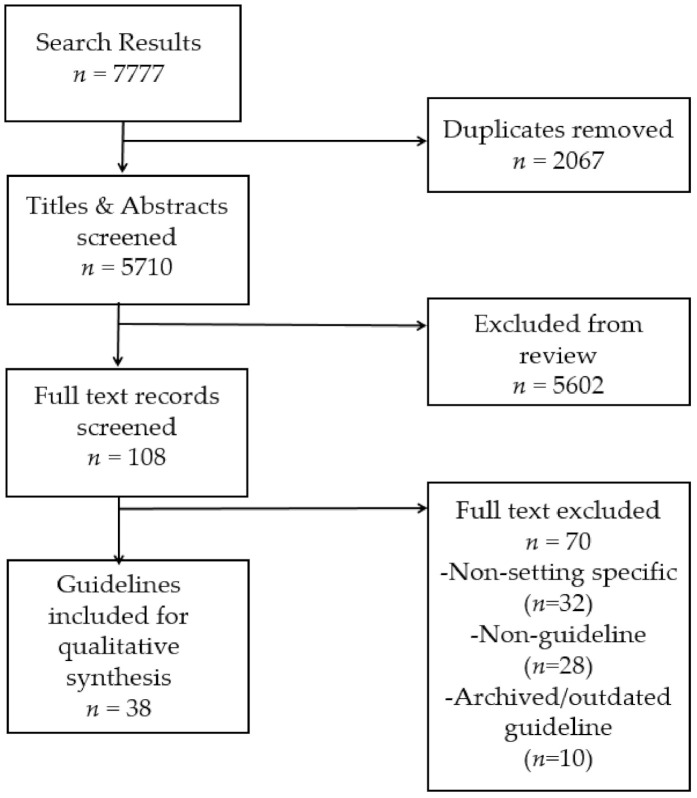
PRISMA flow diagram.

**Table 1 ijerph-18-00838-t001:** Characteristics Table of Included Guidelines.

Guideline	Jurisdiction Area	Publisher	Year Published	Target Group	Overall Quality Assessment	Risk Factors Targeted				
						DB	PA	SB	ST	SLP
Ten ways to empower children to live healthy lives: standards to empower childcare facilities in Arizona (3rd Ed) [48]	Arizona, USA	Arizona Nutrition Network	2016	Children 0–6 years	Recommended with modifications	✓	✓	✓	✓	
Action Plan for Preventing Child and Adolescent Obesity. Promoting Healthy Lifestyles and Preventing Obesity in the Child Care setting [49]	Iowa, USA	The Child and Adolescent Obesity Task Force	1999	Children in child care	Not recommended	✓	✓	✓	✓	
Active Early: a Winconsin guide for improving childhood physical activity (2nd Ed) [42]	Winconsin, USA	Winconsin Department of Health Services	2018	Children 0–6 years	Recommended with modifications		✓	✓	✓	
Healthy Bites: a Winconsin guide for improving childhood nutrition (2nd Ed) [33]	Winconsin, USA	Winconsin Department of Health Services	2019	Children 0–6 years	Recommended with modification	✓				
Australian 24-h movement guidelines for the early years (birth to 5 years) [43]	Australia	Australian Government, Department of Health	2019	Children 0–5 years	Recommended with modifications		✓	✓	✓	✓
Position of the Academy of Nutrition and Dietetics: Benchmarks for Nutrition in Childcare [50]	USA	Academy of Nutrition and Dietetics	2018	Children 2–5 years	Recommended with modifications	✓	✓			
Best Practices for Healthy Eating: a guide to help children grow up healthy [34]	Delaware, USA	Nemours Health and Prevention Service	2008	Children 0–6 years	Recommended with modifications	✓				
Best Practices for Physical Activity: for organizations serving children and youth a guide to help children grow up healthy [44]	Delaware, USA	Nemours Health and Prevention Service	2009	Children 0–6 years	Recommended with modifications		✓	✓	✓	
Canadian 24-h movement guidelines for the early years (0–4 years): an integration of physical activity, sedentary behaviour and sleep [45]	Canada	The Canadian Society for Exercise Physiology	2017	Children 0–4 years	Recommended without modifications		✓	✓	✓	✓
Caring for Children [35]	NSW, Australia	NSW Ministry of Health	2014	Children 0–5 years	Recommended with modifications	✓				
Caring for our children, National Health and Safety Performance Standards: Guidelines for Early Care and Education Programs [51]	Washington, USA	American Academy of Paediatrics, American Public Health Association and National Resource Centre for Health and Safety in Child Care and Early Education	2012	Children 0–6 years	Recommended without modifications	✓	✓		✓	
Early childcare obesity prevention recommendations [52]	USA	Harvard T.H. Chan School of Public Health	-	Children 0–6 years	Recommended with modifications	✓	✓	✓	✓	✓
Early childhood obesity prevention policies [53]	USA	Committee on Obesity Prevention Policies for Young Children, Institute of Medicine	2011	Children 0–5 years	Recommended without modifications	✓	✓	✓	✓	✓
Eat Better Start Better: Voluntary Food and Drink Guidelines for Early Years Setting in England—a Practical Guide [36]	United Kingdom	Action for Children	2017	Children 6 months–4 years	Recommended with modifications	✓				
Eating Well for Under-5s in Child Care: Practical and nutritional guidelines, 2nd edition [54]	United Kingdom	The Caroline Walker Trust	2006	Children 0–5 years	Recommended with modifications	✓	✓			
Enacting eight policies to prevent childhood obesity: projected outcomes for Louisiana [46]	Louisiana, USA	Pennington Biomedical Research Center, Louisiana State University System	2013	Children 1–5 years	Recommended with modifications		✓	✓	✓	✓
Food and Nutrition Guidelines for Pre-school services [55]	Ireland	Health promotion unit department of health and children	2004	Children 0–5 years	Recommended with modifications	✓	✓			
Food & Nutrition for childcare settings: Best Practice guidance [56]	Wales	Llywodraeth Cymru Welsh Government	2018	Children 0–5 years	Recommended with modifications	✓	✓	✓	✓	
Get Up & Grow: Healthy Eating and Physical Activity for Early Childhood [57]	Australia	Australian Government Department of Health and Ageing	2009	Children 0–5 years	Recommended with modifications	✓	✓			
Menu Planning in Childcare [37]	ACT, Australia	ACT Nutrition Support Service	2016	Children 0–5 years	Recommended with modifications	✓				
Long Day Care Menu Planner [38]	NT, Australia	Northern Territory Government	2016	Children 0–4 years	Recommended with modifications	✓				
Start Right, Eat Right [39]	VIC, Australia	State Government of Victoria, Department of Health	2004	Children 2–4 years	Recommended with modifications	✓				
Healthy Food and Drink Guidance- Early Learning Services [40]	New Zealand	Ministry of Health, Manatu Hauora, New Zealand Government	2020	Children 0–6 years	Recommended with modifications	✓				
Healthy Kids, Healthy Future [58]	USA	Nemours, Children’s Health System	-	Children 0–5 years	Recommended with modifications	✓	✓		✓	
Improving healthy weight in children: The healthiest next generation initiative [59]	Washington State, USA	Washington State Department of Health	2014	Children 3–4 years	Recommended with modifications	✓	✓			
Let’s Go! Health Care Toolkit [60]	Portland, Maine, USA	Maine Health	2015	-	Recommended with modifications	✓	✓		✓	✓
Menu planning guidelines for long day care [41]	VIC, Australia	Healthy Eating Advisory Service	2020	Children 1–5 years	Recommended with modifications	✓				
Missouri Move Smart Child Care [47]	Missouri, USA	Missouri Department of Health and Senior Services	2018	Children 0–5 years	Recommended with modifications		✓	✓	✓	
Model Child care health policies (5th Edition) [61]	Pennsylvania, USA	Early childhood education Linkage system: Healthy child care Pennsylvania	2014	Young children in childcare	Recommended with modifications	✓	✓		✓	✓
Much & Move [62]	NSW, Australia	NSW Ministry of Health	2017	Children 0–5 years	Recommended with modifications	✓	✓	✓	✓	✓
Nutrition and Physical Activity Best Practices for Child Care Centers [63]	New York City, USA	NYC Health	2015	Children 0–5 years	Recommended with modifications	✓	✓	✓	✓	✓
Nutritional Guidance for early years food choices for children aged 1–5 years in early education and childcare settings [64]	Edinburgh, Scotland	Scottish Executive	2006	Children 1–5 years	Recommended with modifications	✓	✓			
Nutrition and Wellness Tips for Young Children: provider handbook for the child and adult care food program [65]	Virginia, USA	USDA Food and Nutrition Service	2013	Children 2–5 years	Recommended with modifications	✓	✓		✓	
Right Bite, Easy Guide to Healthy Food and Drink Supply for South Australian Schools and Preschools [66]	SA, Australia	SA Department of Education and Children’s Services and SA Health	2008	Children in pre-school	Recommended with modifications	✓				
Setting the Table- Nutritional Guidance and Food Standards for Early Years Childcare Providers in Scotland [67]	Scotland, UK	NHS: Health Scotland	2018	Children 1–5 years	Recommended with modifications	✓	✓	✓		
Sit less, move more, sleep well: Active play guidelines for under-fives (NZ) [17]	New Zealand	Ministry of Health. Manatu Hauora.	2017	Children 0–5 years	Recommended with modifications		✓	✓	✓	
The preschool initiative: building a healthy foundation for life [68]	Philadelphia, USA	The Food Trust	2011	Children 3–5 years	Recommended with modifications	✓	✓			
Wellness Guidelines for Alaska’s Young Children: A toolkit for child care providers and families [69]	Alaska, USA	State of Alaska, Department of Health and Social Services.	2017	Children 0–5 years	Recommended with modifications	✓	✓	✓	✓	

DB: Dietary Behavior; PA: Physical Activity; SB: Sedentary Behavior; ST: Screen time; SLP: Sleep.

**Table 2 ijerph-18-00838-t002:** Dietary behavior policies and practices within included guidelines (*n* = 38).

ANGELO * Modifiable Environment	Recommended Dietary Behavior Policy and Practice Themes	Frequency of Recommendation
Physical	1. Set Nutrition Standards For The Food And Beverages Available In Early Childhood Education And Care (ECEC) Setting	30 [33,34,35,36,38,39,40,41,48,49,50,51,52,53,54,55,56,57,58,59,60,61,62,63,64,65,66,67,68,69]
	1.1. Nutrition standards, recommended serving sizes for foods, snacks, meals and beverages provided are aligned with national nutrition guidelines	28 [33,34,35,36,38,39,40,41,49,50,51,52,53,54,55,56,57,58,59,60,62,63,64,65,66,67,68,69]
	1.2. Provide a variety of healthy foods from the main food groups in age appropriate portion sizes	28 [33,34,35,36,38,39,40,41,49,50,51,52,53,54,55,56,57,58,59,60,62,63,64,65,66,67,68,69]
	1.3. Specific standards for ECEC food service menus	18 [33,34,35,36,38,39,41,49,51,52,54,56,59,63,65,66,67,69]
	1.4. Water should be widely available to children at all times	23 [33,34,35,36,38,40,41,49,51,54,55,56,57,58,59,61,62,63,64,65,67,68,69]
	1.5. Limit serves and types of sugar sweetened beverages (including fruit juice)	25 [33,34,35,36,38,40,41,48,49,50,51,52,54,55,56,58,59,60,61,63,64,65,67,68,69]
	1.6. Offer age appropriate milk and beverages (i.e., no tea, coffee or energy drinks)	22 [33,34,35,36,38,40,41,52,54,55,56,58,59,60,61,62,63,64,65,67,68,69]
	1.7. Keep high energy, low nutrient foods (e.g., sweets, confectionary, high fat/salty snacks) out of the childcare	10 [33,38,40,52,54,57,61,63,66,67]
Policy	2. Develop And Adopt A Healthy Eating Policy	3 [39,55,59]
	2.1. Nutrition policy is reviewed (annually and/or by an expert i.e., dietitian)	2 [55,59]
	2.2. Parents are involved in the development of the policy	2 [39,55]
	2.3. Ensure staff are willing to working within the policy	1 [39]
	2.4. Provide a copy of the policy to parents and staff	1 [55]
Sociocultural	3. Provide Opportunities For Nutrition Education	11 [33,35,39,49,50,55,57,64,65,68,69]
	3.1. Offer a variety of food awareness/education activities (i.e., allow children to experiment with different foods (shapes, colors, textures), and discuss food preferences and family food traditions)	11 [33,35,39,49,50,55,57,64,65,68,69]
Sociocultural	4. Staff Training In Nutrition Curriculum And Practices	4 [39,49,50,53]
	4.1. Educators are trained in nutrition curriculum/education	1 [39]
	4.2. Educators are trained to understand children can be healthy at a variety of weights and avoid referring to child’s body size	1 [49]
	4.3. Educators are trained in implementing health eating practices	2 [39,50]
	4.4. Food Service Staff are trained food hygiene and safety	1 [39]
Sociocultural	5. Educator Feeding Practices To Encourage Healthy Eating	16 [33,38,39,48,49,50,53,54,55,56,58,59,61,62,63,69]
	5.1. Food is not to be used as reward or punishment	9 [38,48,49,55,56,59,61,63,69]
	5.2. Avoid celebrating special occasions with food or using as a reward	2 [38,63]
	5.3. Don’t force or bribe children to eat	3 [55,63,69]
	5.4. Encourage children to taste different fruit and vegetables each day (praise children for eating healthy foods)	7 [33,54,55,56,61,62,63]
	5.5. Educators should involve children in preparing food and laying and clearing the table	3 [54,61,63]
	5.6. Staff sit with children during meals, and role model healthy behaviors (i.e., eat and drink healthy food and beverages in front of children)	5 [33,49,50,53,69]
	5.7. Educators discuss the food served with children	3 [39,49,58]
Physical	6. Create An Environment That Encourages And Promotes Healthy Eating	25 [33,35,38,39,40,49,50,51,52,53,54,55,56,57,58,59,61,62,63,64,66,67,69]
	6.1. Allow children adequate time to eat	9 [35,39,49,53,54,55,56,61,64]
	6.2. Provide healthy options in appropriate serves, and allow children to self-serve (allow children to choose which foods they eat and how much)	14 [33,35,39,49,53,54,55,57,58,61,62,64,69]
	6.3. Continue to offer children healthy options (repeat exposure and offer equal options)	5 [49,53,55,61,69]
	6.4. Create a relaxed, enjoyable and social meal time environment (i.e., family style mealtimes to encourage child-child and child-educator interactions)	12 [33,39,50,55,56,57,58,59,61,62,64,69]
	6.5. Ensure regular and consistent meal and snack patterns (should be consistent and predictable schedule)	10 [33,35,39,51,53,54,55,57,62,64]
	6.6. Make fruit and vegetables snacks widely available and easily accessible	1 [63]
	6.7. Display healthy eating materials on the walls of the eating room	1 [55]
Sociocultural	7. Parent Engagement	9 [33,39,50,52,53,55,56,57,58,59,69]
	7.1. Encourage parents to pack healthy food from home and ensure foods bought from home meet nutrition written standards	6 [50,52,53,55,57,69]
	7.2. Encourage family involvement in healthy eating at the ECEC e.g., take menu suggestions from parents that are consistent with healthy eating guidelines	6 [33,39,52,53,55,56]
	7.3. Provide a copy of written nutrition guidelines to parents	3 [52,55,69]
	7.4. Provide parents a copy of ECEC menu	4 [52,55,56]
	7.5. Offer parent nutrition education as part of the ECEC program	1 [39]
	7.6. Make parents aware of nutrition learning activities provided to children	1 [39,69]

***** ANGELO: Analysis Grid of Environments Linked to Obesity.

**Table 3 ijerph-18-00838-t003:** Physical activity and sedentary behavior policies and practices within included guidelines (*n* = 38).

ANGELO * Modifiable Environment	Recommended Physical Activity Policy and Practice Themes	Frequency of Recommendation
Physical; Sociocultural	1. Provide Opportunities For Children To Be Physically Active (More Is Better)	28 [17,35,42,43,44,45,46,47,48,49,50,51,52,53,54,55,56,57,59,60,61,62,63,65,67,68,69]
	1.1. Ensure physical activity is incorporated into daily routines and formal childcare curriculum	5 [42,52,53,54,63]
	1.2. Include at least 180 min of physical activity of any intensity, spread throughout the day	16 [42,43,45,48,50,52,53,55,57,59,60,62,65,67,68,69]
	1.3. For children 3–4 years, include at least 60 min of moderate-to-vigorous physical activity during the day	12 [43,45,48,51,52,54,59,61,62,65,68,69]
	1.4. Include opportunities for adult-led, structured physical activity	10 [42,44,47,48,51,53,59,63,65,69]
	1.5. Include opportunities for unstructured physical activity, free play (play-time)	9 [42,44,46,47,48,49,63,65,69]
	1.6. Provide daily opportunities for activity through outdoor playtime (should be supervised)	11 [17,42,44,49,51,53,55,56,61,65,69]
	1.7. Provide opportunities for children to develop and practice gross motor and movement skills	4 [49,51,62,63]
	1.8. Include culturally appropriate physical activities	1 [63]
Policy; Sociocultural	2. Develop And Adopt Policies For Physical Activity And Physical Activity Education Programs	3 [33,53,69]
	2.1. Engage staff and parent support for physical activity standards	1 [53]
	2.2. Seek consultation from experts annually on the physical activity programs delivered in the childcare	1 [53]
	2.3. Provide parent education at least 2 times a year (to reduce screen time)	1 [33]
	2.4. Develop a written policy promoting physical activity and the removal of barriers to physical activity participation (including limiting screen time)	1 [69]
Sociocultural	3. Offer Educator Training To Provide Safe And Developmentally Appropriate Physical Activity	2 [53,69]
	3.1. Staff should be trained to provide guidance to parents to encourage physical activity	1 [53]
	3.2. Staff should be trained to provide guidance to parents in appropriate sleep duration	1 [53]
	3.3. Staff should be trained in encouraging child physical activity and decreasing sedentary behavior	1 [53]
	3.4. Offer staff annual training opportunities in physical activity programs and practices	1 [69]
Sociocultural	4. Educators To Promote The Benefits Of Physical Activity With Children	8 [44,49,53,59,63,65,68,69]
	4.1. Educators should model physical activity by participating in activities	5 [17,49,53,63,69]
	4.2. Engage children in physical activity they enjoy, including games and sport (age appropriate, fun and offer variety)	5 [17,44,49,63,68]
	4.3. Expressive play is encouraged e.g., music, dancing and make believe	3 [17,49,63]
	4.4. Educators embed physical activity into educational activities	2 [49,53]
	4.5. Avoid punishing children for being physical active	1 [53]
	4.6. Avoid withholding physical activity as a punishment	4 [53,59,63,69]
	4.7. Elimination games should be avoided as well as competitive activates and games	3 [43,63,65]
	4.8. Engage equal participation from boys and girls in physical activity	1 [63]
	4.9. Celebrate special occasions with physical activity (games, dancing and extra playground time).	1 [63]
Physical; Sociocultural	5. Limit The Time Children Spend Sitting (Less Is Best)	14 [17,42,43,44,45,47,48,49,52,53,57,62,63,67]
	5.1. Children should not be sitting for extended periods (or be restrained) for more than 30–60 min at a time	12 [42,43,44,45,47,48,49,52,53,57,62,63]
	5.2. When sedentary, children should be engaged in educational and creative pursuits, and be engaged socially.	3 [43,45,62]
	5.3. Engage children that tend to be sedentary in active play	2 [45,49]
Physical; Sociocultural	6. Limit The Use Of Screen Time (Less Is Best)	23 [17,33,42,43,44,45,46,47,48,49,51,52,53,56,57,59,60,61,62,63,65,67,69]
	6.1. No screen time is recommended for children <2 years	16 [17,43,45,46,47,48,51,53,56,57,59,60,61,62,65,69]
	6.2. No more than 1 h of screen time/week is recommended for children aged 2 or above	18 [17,43,44,45,46,47,49,51,52,53,56,57,59,60,61,62,63,65]
	6.3. Screens should not be used/available during mealtimes or nap times	5 [48,51,61,65,69]
	6.4. Limit the use of screen time for educational activities or active movement programs	5 [42,47,61,65,69]
	6.5. Parent permission should be requested for children to participate in any screen based activity	1 [44]
	6.6. Screen time should be supervised by an adult (to help children apply what they are learning)	2 [44,51]
	6.7. When offered, screen/digital media should be free from advertising, violence or should that tempt children to overuse	4 [51,61,63,65]
	6.8. Work with parents to limit overall screen time	1 [33]
Physical	7. Support Healthy Sleeping Habits	9 [17,43,45,52,53,56,60,61,62]
	7.1. Include a nap within the daily routine, with regular sleep and wake-up times	6 [17,43,45,52,61,62]
	7.2. Provide an environment that provides restful sleep: remove screen media from sleeping/napping areas and low noise	4 [52,53,56,61]
	7.3. Maintain a calm nap-time routine	2 [52,53]
Physical	8. Create A Physical Environment That Promotes Physical Activity	6 [44,49,53,54,63,69]
	8.1. Provide play equipment that encourages physical activity	4 [44,49,53,54]
	8.2. Provide simple play equipment to encourage creative play and exploration (e.g., cardboard boxes) and portable play equipment that encourages indoor and outdoor play	4 [49,53,54,63]
	8.3. Provide adequate space for children to be physically active	4 [44,53,54,69]
	8.4. Ensure the outdoor area offers variety in terms of secure equipment in shade, open grass and varying surfaces	2 [53,54]
	8.5. Ensure that the educator to child ratio is fairly low (i.e., less than 10 children to one educator)	1 [63]

***** ANGELO: Analysis Grid of Environments Linked to Obesity.

## Data Availability

No new data were created or analyzed in this study. Data sharing is not applicable to this article.

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
