# Peer review of "Obesity Prevention within the Early Childhood Education and Care Setting: A Systematic Review of Dietary Behavior and Physical Activity Policies and Guidelines in High Income Countries"

_ijerph, 2021, doi:10.3390/ijerph18020838_

Round 1
Reviewer 1 Report
This manuscript was a systematic narrative review of policies related to early education centers and policies related to reducing obesity. Overall, the authors were able to adequately synthesize the context of these policies, however, the reviewer has provided additional comments to further strengthen this manuscript.
Title: Understanding there may be a certain character count for this journal, this may not be applicable, but consider adding nutrition policies or physical activity policies as right now it is open to interpretation. Also, consider including if this was a focus globally or just in Europe, USA, etc.
Abstract: Recommend commenting on the problem with all these policies, that globally childhood obesity continues to rise, so exploring why these policies may not be effective…. Explain how many databases were used to identify these articles. Explain if the PRISMA method was used. Also, this manuscript could be construed for more of a narrative review as even though a systematic approach was used, since this appeared to be more exploratory in nature. For the conclusive statement, this is certainly true, but overall, how will this help reduce childhood obesity? Maybe consider another way to conclude this abstract.
Introduction:
Even though readers of this journal may know how infant/childhood obesity is defined, it may be good to include that information as one may use CDC whereas someone else may use WHO’s definition.
Line 58: if there could be further explanation about eating behaviors and all the components that are included within this context, it would help readers who are not familiar with the difference between eating and dietary behaviors.
Methodology:
Recommend that exclusion criterion would also be those articles that were studies as this was just a scoping review on the policies, not necessarily the outcomes (e.g. reduction in obesity). Specify earlier in the methodology that the PRISMA method was used to guide the authors.
Identify the key words used as it is not apparent.
Results:
Mentioned that outdated policies were excluded, clarify what this means within your methodology.
For the quality scoring, if some articles score poorly, what was the rationale for still including it within the analysis? Please clarify this in the methodology.
For the results, in the sub-headings clarify if referring to eating behaviors or physical activity.
Discussion:
There was limited discussion on what all this means. It was a synthesis of the results, but limited information about what this actually means in the context of obesity and how this actually helps. So, there could have been more information connecting these policies to actual studies. A few were mentioned, but limited discussion or further elaboration on them.
Author Response
Point 1: This manuscript was a systematic narrative review of policies related to early education centers and policies related to reducing obesity. Overall, the authors were able to adequately synthesize the context of these policies, however, the reviewer has provided additional comments to further strengthen this manuscript.
Response 1: We would like to thank this reviewer for providing considered and thoughtful comments on this manuscript. Please find we have attempted to address each comment below.
Point 2: Title: Understanding there may be a certain character count for this journal, this may not be applicable, but consider adding nutrition policies or physical activity policies as right now it is open to interpretation. Also, consider including if this was a focus globally or just in Europe, USA, etc.
Response 2: Thank you for this comment. Please find we have extended our title to “Obesity prevention within the early childhood education and care setting: a systematic review of dietary behaviour and physical activity policies and guidelines in high income countries”.
Point 3: Abstract: Recommend commenting on the problem with all these policies, that globally childhood obesity continues to rise, so exploring why these policies may not be effective…. Explain how many databases were used to identify these articles. Explain if the PRISMA method was used. Also, this manuscript could be construed for more of a narrative review as even though a systematic approach was used, since this appeared to be more exploratory in nature. For the conclusive statement, this is certainly true, but overall, how will this help reduce childhood obesity? Maybe consider another way to conclude this abstract.
Response 3: The reviewer has made some great suggestions to improve the clarity of our abstract. However given that the journal specifies that the abstract should be a total of about 200 words maximum, only some of these points were able to be further elaborated within the abstract. Further, the authors believe that concluding statement adequately aligns with journal guidelines, which are to “indicate the main conclusions or interpretations”, however some relevant detail has been added to help strengthen our conclusion. The abstract now reads as:
“As a strategy for early childhood obesity prevention, a variety of dietary behavior and physical activity policies and guidelines published by leading health agencies and Early Childhood Education and Care (ECEC) licensing and accreditation bodies exist. Given the potential diversity in recommendations from these policies, this narrative review sought to synthesize, appraise and describe the various policies and guidelines made by organizational and professional bodies to highlight consistent recommendations and identify opportunities to strengthen such policies. An electronic bibliographic search of seven online databases and grey literature sources was undertaken. Records were included if they were policies or guidelines with specific recommendations addressing dietary behavior and/or physical activity practice implementation within the ECEC setting; included children aged >12 mo and <6 yr, and were developed for high income countries. Recommended dietary behavior and physical activity policies and practices were synthesized into broad themes using the Analysis Grid for Environments Linked to Obesity framework, and the quality of included guidelines appraised. Our search identified 38 eligible publications mostly from the US and Australia. Identified guidelines were largely consistent in their recommendation and frequently addressed the physical and sociocultural environment, and were well-aligned with research evidence. Broader consideration of policy and economic environments may be needed to increase the impact of such policies and guidelines within the ECEC setting.”
Point 4: Introduction: Even though readers of this journal may know how infant/childhood obesity is defined, it may be good to include that information as one may use CDC whereas someone else may use WHO’s definition.
Response 4: The authors thank the reviewer for this suggestion, we have added a details as to how infant/childhood obesity is defined.
Introduction, page 1: “Childhood overweight and obesity is increasingly prevalent, and if global trends continue, will affect up to 70 million infants and young children by the year 2025 [1]. Childhood obesity is defined by an excess of body fatness that is widely categorized according to body mass index scores adjusted for child sex and gender [1]. Given that childhood obesity can track throughout the lifespan and influence lifelong health trajectories [2], it has been identified as one of the most serious public health challenges of the 21st Century [1].”
Point 5: Line 58: if there could be further explanation about eating behaviors and all the components that are included within this context, it would help readers who are not familiar with the difference between eating and dietary behaviors.
Response 5: The authors would like to thank this reviewer for identifying this point. After exploring the various definitions of these terms (as per Dietary Behavior: An Interdisciplinary Conceptual Analysis and Taxonomy (nih.gov)), it is clear that the use of the term ‘eating behaviours’ (i.e. a term that is focused around eating habits, eating occasions and portion sizes) has been used in error, and instead should read ‘dietary behaviours’ to encompass more broadly food choices (i.e. preferences and intensions), eating behaviours and dietary intake. Please find the use of this term has been amended within the manuscript and now includes a reference to Stok et al. outing a taxonomy for defining dietary behavior. Additionally, a brief definition of all the components that are included under “dietary behaviors” has also been added:
Methods, page 3: “This review sought to include ECEC focused recommendations and guidelines that aimed to prevent obesity in preschool aged children and targeted specific obesity prevention behaviors including dietary behaviors (inclusive of guidelines influencing food choices (i.e. preferences and intentions), eating behaviors (i.e. eating habits, eating occasions and portion sizes), and dietary intake/nutrition (i.e. food and nutritional make-up on intakes and diets) [3]), physical activity, sedentary behavior/screen time and/or sleep”.
Please find the term dietary behaviours has also been used consistently throughout the manuscript.
Point 6: Methodology: Recommend that exclusion criterion would also be those articles that were studies as this was just a scoping review on the policies, not necessarily the outcomes (e.g. reduction in obesity).
Response 6: The following has been specified with the methodology:
Methods, page 3: “ECEC based practices developed as part of obesity prevention interventions/programs and evaluated in conventional trials (e.g. Nutrition and Physical Activity Self-Assessment for Child Care (NAPSACC)) [20], were not included in this definition”.
Point 7: Specify earlier in the methodology that the PRISMA method was used to guide the authors.
Response 7: Pease note this detail has been added to the manuscript.
Methods, page 3: “This review has been conducted and reported in accordance to the PRISMA guidelines [2]. A protocol or registration of this review has not been previously published”.
Point 8: Identify the key words used as it is not apparent
Response 8: Please find this detail has now been explicitly added in text, to include:
Methods, page 4:“The search combined the following search terms: (1) ECEC and, (2) Food/Nutrition or physical activity or screen time or sedentary behavior or sleep or obesity, and (3) Policy/guidelines (Medline Search Strategy is available in Table S1)”
Point 9: Results: Mentioned that outdated policies were excluded, clarify what this means within your methodology
Response 9: The following is reported in the manuscript:
Methods, page 4: “If one advisory body or organization had published more than one guideline on the same subject, then the most recent update was selected for inclusion. Thus, only findings from the latest version of published guidelines were included in the synthesis.”
Point 10: For the quality scoring, if some articles score poorly, what was the rationale for still including it within the analysis? Please clarify this in the methodology.
Response 10: The aims of the review were to provide a comprehensive review of the available nutrition and PA policies and practices for the sector. This was to provide a broad overview of the types of recommendations for all countries of interest. The exclusion of policies that were lower quality would have precluded the inclusion of recommendations for a number of countries. As such, the synthesis included findings from all included policies and practices, regardless of study quality. However, for countries without any existing policies, or those seeking to strengthen their policies, we recommend that the higher quality guidelines form the basis of their recommendations. Please find some rationale has been added.
Discussion, page 21: “Our assessment of guideline quality using the AGREE-2-tool identified three guidelines that scored highly on the tool and were recommended without modification. Thus, given that very few included guidelines scored high quality ratings, our synthesis has included findings from all identified policies and guidelines regardless of quality, as exclusion of guidelines due to low quality ratings would have precluded the inclusion of recommendations from a number of countries. However, it is possible that the guidelines identified as high quality could be adopted or be used as a reference document for jurisdictions that do not have existing ECEC guidelines, provided they are contextually relevant and appropriate.”
Point 11: For the results, in the sub-headings clarify if referring to eating behaviors or physical activity
Response 11: Please find this detail has been added to the sub-headings in the results section to support clarification.
Point 12: Discussion: There was limited discussion on what all this means. It was a synthesis of the results, but limited information about what this actually means in the context of obesity and how this actually helps. So, there could have been more information connecting these policies to actual studies. A few were mentioned, but limited discussion or further elaboration on them.
Response 12: Thank you for this comment. We have expanded in the discussion section alignment of each of the recommendations to existing evidence and discuss the potential implications of this for obesity prevention more broadly. Please note the following additions/changes have been made to the discussion:
Page 20: “In order for ECEC guidelines and policies to have an impact on child health outcomes, they need to be supported by empirical evidence. Our review identified that the most consistent recommendations were strongly supported by empirical evidence.”
Page 21: “Additionally, there were few recommendations around tailoring dietary behavior and physical activity opportunities to be more culturally appropriate. Given the broad reach and diversity of children attending such settings, future guidelines should consider the inclusion of culturally competent guidance, as cultural and religious preferences are increasingly acknowledged as an important mediator for child access, acceptability and preferences for certain health behaviors known to influence child health outcomes [8].”
Page 21: “It was reassuring to find that ECEC based obesity prevention policies and guidelines aligned with current research evidence, but highlights the need for ongoing investment to support policy implementation. Even with the introduction and existence of such policies in a number of countries in the last decade, alarmingly the rates of childhood obesity have continued to increase [9, 10]. While many studies have shown that despite the availability of such guidance, few ECEC services implement these evidenced-based recommendations [11, 12]. Thus, while there remain opportunities to strengthen current guidelines, the impact of such policies cannot be achieved without the systematic development of dissemination and implementation strategies to facilitate their successful update [13, 14]. Additionally, the introduction of macro-level incentives and strategies such as their inclusion in regulatory standards, may be useful to facilitate the wide-spread adoption of such recommendations within the setting."
Page 22 (conclusion): “This review provides an overview of the recommendations and identifies high quality ECEC based policies and guidelines could form as the basis for developing future childhood obesity prevention guidance or unified recommendations for the sector.”
Reviewer 2 Report
This manuscript entitled "Obesity prevention within the early childhood education and care setting: a systematic review of policies and guidelines" aimed to identify and describe the various policies and guidelines made by organisational and professional bodies.
This is a very interesting study. However, some issues should be addressed by the authors before publication, as follows:
Abstract
- Aim is not clear and should be rewritten;
- Please include the databases analysed;
- Results can be explore more in-depth;
- Lacks from a conclusion.
Introduction
- Why does the authors restrict only in high income countries? This rationale should be clear in the introduction section as well as in the inclusion criteria in the methods section.
Methods
- Preschool age needs a reference;
- Lines 104 to 107 do not support the reason to choose only high income countries.
- Although some characteristics were showed in the introduction, it is not clear what is the ANGELO framework in the methods section and how the authors have managed it.
References
- Recent references from Int. J. Environ. Res. Public Health can be included in the introduction and discussion sections.
Author Response
Point 1: This manuscript entitled "Obesity prevention within the early childhood education and care setting: a systematic review of policies and guidelines" aimed to identify and describe the various policies and guidelines made by organisational and professional bodies. This is a very interesting study. However, some issues should be addressed by the authors before publication, as follows:
Response 1: The authors would like to thank this reviewer for their timely comment on this manuscript. We were very pleased to hear the reviewer found this research of interest, and have made an effort to address the issues highlighted below.
Point 2: Abstract: Aim is not clear and should be rewritten; Please include the databases analysed; Results can be explore more in-depth; Lacks from a conclusion
Response 2: Please note detail has been added to the abstracts aim to improve clarity: “Given the potential diversity in recommendations from these policies, this narrative review sought to synthesize, appraise and describe the various policies and guidelines made by organizational and professional bodies to highlight consistent recommendations and identify opportunities to strengthen such policies”;
Detail regarding the number of databases searched was also added: “An electronic bibliographic search of seven online databases and grey literature sources was undertaken”;
Additional amendments have been made to the abstract in line with reviewer one, however given that the journal specifies that the abstract should be a total of about 200 words maximum, only some of these points were able to be further elaborated within the abstract. Further, the authors believe that results and concluding statement adequately aligns with journal guidelines.
Point 3: Introduction: Why does the authors restrict only in high income countries? This rationale should be clear in the introduction section as well as in the inclusion criteria in the methods section.
Response 3: Please note that such detail has been added to the introduction and methods.
Introduction, page 3: “However, the structural, organizational and funding models applied to the ECEC setting can vary substantially between high and low income countries, which is likely to influence the available infrastructure and political support for obesity prevention initiatives in this setting [3]. For this reasons, this review sought to examine only the policies and guidelines in high income countries to ensure context consistency, comparability and relevance.”
Point 4: Methods: Preschool age needs a reference;
Response 4: Please find the following reference has been added to the methods (page 3): OECD- Social Policy Division - Directorate of Employment Labour and Soical Affairs, PF3.2: Enrolement in childcare and pre-school, in Public policies for families and children. 2019: OECD Family Database. Access at: PF11: (oecd.org)
Point 5: Methods: Lines 104 to 107 do not support the reason to choose only high income countries.
Response 5: Please find additional support for this decision has been added to the methods, page 3:
“Given the wide variation in cultural, social, ecological and political contexts internationally (which can influence how communities and governments prioritize early childhood obesity prevention) [4], this review focused on both regional and national guidelines from countries ranked within the OECD top 20 countries for highest average annual incomes for 2019 [19].”
Point 6: Although some characteristics were showed in the introduction, it is not clear what is the ANGELO framework in the methods section and how the authors have managed it.
Response 6: Please find some additional detail has been added to the methods (page 5): “The identified themes were then mapped to the ANGELO framework category definitions to provide an overview of where the primary recommendations focus and to guide our narrative summary of study results”.
Point 7: Recent references from Int. J. Environ. Res. Public Health can be included in the introduction and discussion sections.
Response 7: Thank you for highlighting this point. Please find references to the following Int. J. Environ. Res. Public Health journal articles have been added to support our rationale and discussion:
Makanjana, O. and A. Naicker, Nutritional Status of Children 24–60 Months Attending Early Child Development Centres in a Semi-Rural Community in South Africa. International Journal of Environmental Research and Public Health, 2021. 18(1): p. 261.
Spence, A., et al., Childcare Food Provision Recommendations Vary across Australia: Jurisdictional Comparison and Nutrition Expert Perspectives. International Journal of Environmental Research and Public Health, 2020. 17(18): p. 6793
Reviewer 3 Report
Thank you for asking me to review this manuscript.
A lot of attention and resources are spent on lifestyle medicine in adults, but it's application in children receives significantly less attention. However, children, at increasingly younger ages, are experiencing many of the same lifestyle related chronic conditions as adults.
Paediatric lifestyle medicine encompasses key aspects of preventive medicine, which is at the heart of paediatric care. Lifestyle choices are important determinants of health outcomes. In my opinion this manuscript raises one of the most important public health problem - obesity prevention.
I find the manuscript interesting for the community, but there are some matters that have to be addressed. Please see my comments below.
- “All identified titles and abstracts were screened by a single reviewer against the inclusion criteria” – Why did authors decide that only single reviewer screened 5,710 abstracts? Why not use the same method as full-text screening?
- Authors focused on both regional and national guidelines from countries raked within the OECD top 20 countries for highest average annual incomes. Please complete information about the year from which these data come? It’s a top 20 countries for highest average annual incomes in 2019?
Author Response
Point 1: Thank you for asking me to review this manuscript. A lot of attention and resources are spent on lifestyle medicine in adults, but it's application in children receives significantly less attention. However, children, at increasingly younger ages, are experiencing many of the same lifestyle related chronic conditions as adults. Paediatric lifestyle medicine encompasses key aspects of preventive medicine, which is at the heart of paediatric care. Lifestyle choices are important determinants of health outcomes. In my opinion this manuscript raises one of the most important public health problem - obesity prevention. I find the manuscript interesting for the community, but there are some matters that have to be addressed.
Response 1: The authors would like to thank this reviewer for providing such an eloquent summary of the field, and for providing our manuscript with such timely and considered comments. We were very pleased to hear the reviewer found this research of interest to the community. Please find our response to reviewer comments below.
Point 2: “All identified titles and abstracts were screened by a single reviewer against the inclusion criteria” – Why did authors decide that only single reviewer screened 5,710 abstracts? Why not use the same method as full-text screening?
Response 2: Given the large number of titles and abstracts returned by our search, the decision to have only one reviewer screen was a pragmatic decision due to resource constraints. It is worth noting however, that this task was conducted by an experienced reviewer, and applied a conservative approach for sending records to full-text screening.
Further, while the Cochrane handbook specifies that “screening of titles and abstracts to remove irrelevant reports should be done in duplicate by two people working independently…it is acceptable that this initial screening of titles and abstracts is undertaken by only one person. It is essential, however that two people working independently are used to make a final determination as to whether each study considered possibly eligible after title/abstract screening meets the eligibility criteria based on the full text of the study report(s). [Chapter 4: Searching for and selecting studies | Cochrane Training]
Point 3: Authors focused on both regional and national guidelines from countries raked within the OECD top 20 countries for highest average annual incomes. Please complete information about the year from which these data come? It’s a top 20 countries for highest average annual incomes in 2019?
Response 3: Please find we included countries from the OECD top 20 countries for highest average annual income for 2019, and this detail has now been explicitly stated in the methodology (page 3). “this review focused on both regional and national guidelines from countries raked within the OECD top 20 countries for highest average annual incomes for 2019”.
Round 2
Reviewer 1 Report
This manuscript has been extensively revised and therefore, reads very well now as it has addressed all comments presented by the reviewer. Even though the English has been checked for minor editing, there did not appear to be any issues in the manuscript, but no other indicator is available for English is fine.